# *Cystoseira compressa* and *Ericaria mediterranea*: Effective Bioindicators for Heavy- and Semi-Metal Monitoring in Marine Environments with Rocky Substrates

**DOI:** 10.3390/plants13040530

**Published:** 2024-02-15

**Authors:** Ilaria Pagana, Vincenzo Nava, Giuseppe D. Puglia, Claudia Genovese, Giuseppe Emma, Carla Salonia, Nicola Cicero, Giuseppina Alongi

**Affiliations:** 1Department of Biological, Geological and Environmental Sciences, University of Catania, 95128 Catania, Italy; ilaria.pagana@phd.unict.it (I.P.); giuseppe.emma@phd.unict.it (G.E.); carlasalonia29@gmail.com (C.S.); g.alongi@unict.it (G.A.); 2Consorzio Nazionale Interuniversitario per le Scienze Del Mare, 00196 Rome, Italy; 3Department of Biomedical and Dental Sciences and Morphofunctional Imaging, University of Messina, 98168 Messina, Italy; vincenzo.nava@unime.it (V.N.);; 4Institute for Agriculture and Forestry Systems in the Mediterranean, National Research Council of Italy, Via Empedocle, 58, 95128 Catania, Italy

**Keywords:** brown algae, Mediterranean Sea, biomonitors, short- and long-term accumulation, Phaeophyceae

## Abstract

Marine environmental monitoring is essential to ensure that heavy-metal (HM) concentrations remain within safe limits. Most seawater analyses currently consider sediment or water samples, but this approach does not apply to rocky substrates, where water samples can only indicate immediate contamination. We used two common Mediterranean algae species, *Cystoseira compressa* and *Ericaria mediterranea*, as bioindicators living in the intertidal zone on rocky substrates along the seacoast. HM concentrations were assessed over a one-year period in the perennial base crust and in the seasonal frond, considering marine sites characterised by different contamination risks. Both algae showed that HMs accumulate mainly in the perennial base rather than in the seasonal frond. Furthermore, the algae species always showed a different order of bioaccumulation factors: Cd > Ni > Pb > Cr > Cu > Mn > Zn for the frond and Pb > Cr > Ni > Cd > Mn > Cu > Zn for the base. Our study shows that *C. compressa* and *E. mediterranea* accumulate HM consistently with the types of sites analysed and differentially with respect to the part of the thallus. These results demonstrate that these algae can be effectively used as reliable bioindicators to assess the presence of HM in marine environments with rocky substrates, providing both short- and long-term monitoring.

## 1. Introduction

Heavy- and semi-metals (HMs) have always been in the environment in small amounts via natural processes [1,2,3]. But, recently, more industrialisation and economic growth along the coasts have raised the levels of HMs in the soil and water, which threatens the environment and marine life [4,5,6]. Indeed, HMs can accumulate in biota, causing toxicological effects and ecosystem imbalances [7,8,9,10,11,12,13]. Therefore, it is crucial to monitor the marine environment to ensure that these levels do not rise above the safe threshold. To date, water columns and sediments are usually used as test samples for the quantification of HM pollution in the marine environment. However, the water analysis can be misleading because the metal concentrations in the solution are often close to the analytical detection limits, and they can be very variable over time [14]. In order to detect significant differences, the water analysis should take into consideration a well-planned sampling programme, which greatly impacts the time and costs of the investigation [14,15]. Sediment analysis, on the other hand, is more effective since HM concentrations remain more stable in this abiotic matrix over time, but this approach is not feasible when hard substrates are present (rocky substrates) or when the seabed is too deep. Furthermore, from the sediment analysis, it is not possible to reconstruct the accumulation trend over time, as the concentration of HMs can change throughout the year and over the years. Recently, biota has also been suggested as a good indicator, i.e., bioindicator, of metal pollution, as the biotic matrix can store contaminants in its tissues, and this can be correlated to a specific period of time [16,17]. Among bioindicators, due to their sensitivity to the local water composition, sessile organisms, especially algae, can serve as a reliable proxy for the degree of contamination in a given area. In fact, they can absorb toxic elements, are easy to sample, and have no trophic relationship with the substrate [18]. In particular, brown algae can accumulate huge amounts of HMs, and this makes them the preferred choice to monitor pollution in coastal areas [19,20,21]. That also aligns with the EU-Marine Strategy Framework Directive (MSFD 2008/56/EC) [22], which recognises these algae as valuable indicators of water and ecosystem quality [23].

In this regard, several studies have already examined the impacts of HMs on species belonging to the *Cystoseira* s.l. [24,25,26,27], highlighting the ability of these algae to accumulate specific HMs [14,20,26,28,29,30,31].

The species of *Cystoseira* s.l. live in the infralittoral zone, mainly on a rocky substrate. Among them, the intertidal species are attached to the substrate by a robust perennial basal crust, while the frond has a seasonal turnover. *Cystoseira compressa* (Esper) Gerloff & Nizamuddin is a widespread species with a large ecological value. This species is mainly found in the Mediterranean Sea and on some Atlantic islands such as the Azores, Bermuda, the Canary Islands, the Cape Verde Islands, and Madeira [32], whereas *Ericaria mediterranea* (Sauvageau) Molinari & Guiry is endemic to the Mediterranean Sea [32] and is most affected by environmental impacts. Therefore, both species of *Cystoseira* s.l., due to their wide distribution, the presence of perennial and seasonal portions, and the ease of being sampled on rocky substrates in the intertidal zone, can represent a valid bioindicator in the entire Mediterranean basin. Thus, the aim of the present study was to assess the use of *C. compressa* and *E. mediterranea* as a biotic matrix to quantify HM pollution in the absence of sediment, as well as to evaluate whether there is a different accumulation capacity between the basal perennial portion and the annual fronds. This approach can pave the way for their use as a reference for long-term and short-term bioaccumulation assessments.

## 2. Results

The concentration of all HMs in the water was always under both the quality standards provided by the U.S. Environmental Protection Agency (EPA) [33] and the European Directive 2013/39/UE [34] regarding annual means and admissible maximum concentration (Table 1). As regards Mn, there are no references to its admissible maximum concentration in seawater.

The yearly average amount of each element in the base and frond of the two species of *Cystoseira* s.l. is shown in Figure 1 and Figure 2 as bar plots and in Appendix A.

The arsenic (As) concentration in the water was always below the detection limit (<LOQ), whereas it was found in the two algae. Particularly, the lowest concentration was detected in *C. compressa* from CM (3.09 mg/kg) and the highest in *E. mediterranea* from PM (10.18 mg/kg). Cadmium (Cd) levels were <0.026 mg/kg on average in water samples of CM and PM and <LOQ in other sites; in the algal portions, it accumulated to the greatest extent in the base of *E. mediterranea* sampled at PM (0.62 mg/kg). Chromium (Cr) was found at low concentrations in the water (≤0.014 mg/kg), and the highest mean values were observed in *E. mediterranea* from PM (1.97 mg/kg), while *C. compressa* exhibited lower accumulation, with the highest in the base from PM (1.75 mg/kg). Copper (Cu) was always found in water, although the amounts varied a lot from sample to sample (highest 0.993 mg/kg in CM and lowest 0.038 mg/kg in PM). Regarding accumulation by algae, it was similar in the fronds of *C. compressa* from CM and PM (0.24 mg/kg; 0.25 mg/kg), as well as in the base of *C. compressa* and *E. mediterranea* from PM (0.5 mg/kg; 0.45 mg/kg); however, the highest and lowest values are found in the base of *C. compressa*, 0.5 in PM and in the base of *E. mediterranea,* 0.02 in Br. Mercury (Hg) accumulation was below the detectable limit in water samples; nevertheless, although in low concentrations, it is found in the algae, particularly in the base of *C. compressa* and *E. mediterranea* from PM and in the frond of *E. mediterranea* from Br, in all cases on average lower than 0.01 mg/kg. Manganese (Mn) was always found in water, with a value of ≤0.985 mg/kg. In the algae, mean levels were highest in the bases of both algae species that were studied in PM, where they ranged from 6.77 mg/kg for *E. mediterranea* to 4.74 mg/kg for *C. compressa*. Manganese levels were much lower in the frond except for *E. mediterranea* from Br, in which the frond has higher Mn values than the base. Nickel (Ni) levels in the water were very low (≤0.039 mg/kg), but they rose sharply in the bases of both species of algae, reaching 3.75 mg/kg for *C. compressa* and 4.35 mg/kg for *E. mediterranea* when samples were collected at PM. Lead (Pb) in water has always been found in low percentages (≤0.012), while in the algae, it was significantly more concentrated, particularly in the base of PM samples, with 1.77 mg/kg for *C. compressa* and 1.67 mg/kg for *E. mediterranea.* Among the analysed elements, the concentration of zinc (Zn) was the highest in the water (≤3.218 mg/kg), even though it was within the standard quality limits. In the algal portions, it accumulated the most in the base of PM samples, ranging from 2.03 to 3.11 mg/kg in *C. compressa* and *E. mediterranea*, respectively.

According to the BAF data shown in Table 2, the bases of *C. compressa* and *E. mediterranea* have the highest capacity to accumulate metals, while the fronds of the two algae exhibit lower ratios. In terms of metals, the BAF decreased with the same trend in the bases of the two algae, in the order Pb > Cr > Ni > Cd > Mn > Cu > Zn, while for the fronds, it followed a different order: Cd > Ni > Pb > Cr > Cu > Mn > Zn. Although the highest BAF value of Cd was found in the fronds, this value was still lower than the corresponding values in the bases. Their absence (<LOQ) in water samples had an impact on the calculation of BAF for As and Hg. The Zn bioaccumulation factor was the lowest, with values lower than 1 in fronds.

The PCA shows the relationships between sampling sites and different portions of the two algae species, with the first two coordinates explaining 88.2% and 91.6% of the total variability for *C. compressa* and *E. mediterranea*, respectively (Figure 3 and Appendix A). As for *C. compressa*, the bases from PM formed a clear cluster with respect to the frond collected from the same site and all algal portions from CM. For this species, the metals providing the highest contribution to the PCA were Cr and As for the first dimension, while for the second dimension, Hg contributed the most. For *E. mediterranea*, regardless of the considered thallus portions, the samples from the control site (Br) grouped together very compactly, while bases and fronds from PM formed two different clusters that were more loosely related. Nevertheless, PCA of *E. mediterranea* samples provides a sharp differentiation of samples on their site source and on the algal portion, shedding light on the different metal accumulation capacities of base and frond in the case of algae sampled at the industrial site (PM). For this PCA, Mn contributed the most for the first dimension, while for the second dimension, Cu was the highest contributor, and it characterised the frond samples from the PM.

## 3. Discussions

To protect marine biodiversity from the risk of HMs pollution, reliable and low-cost bioindicators are needed, especially in highly industrial areas that are more exposed to rapid changes in water chemical composition. In the present study, we have analysed the HM concentrations in water and different algal portions (base and frond) sampled in highly industrial and nonindustrial areas. The results showed that HMs in water samples were always under both European and USA quality standards, but the analyses of algae revealed a consistent accumulation trend between the industrial and control areas. These findings demonstrate that both *Cystoseira* s.l. species can be used as sensitive bioindicators for monitoring seawater composition in areas affected by high levels of industrialisation. The As concentrations, for example, were very low in water samples, regardless of the site source, whereas the algae As accumulation was quite high and significantly correlated to the industrial site. This different way of accumulating As between the abiotic and biotic matrix was also detected by Kut et al. [28], who found lower As concentrations in the sediment samples than in *Gongolaria barbata* (Stackhouse) Kuntze, belonging to the *Cystoseira* complex. Taylor and Jackson [35] demonstrated an active uptake of As in brown algae, which can be used as a catalyst in various physiological processes instead of P. In any case, this accumulation in algae can serve as evidence for the presence of As in a comparative investigation over time between industrial and nonindustrial sites. Furthermore, a similar pattern of As accumulation was also described in three species of the genus *Cystoseira* by Sales et al. [25], who also reported a strong link between the amount of Pb, Cu, and Zn in the macroalgae and site pollution. Previous investigations on macroalgae as bioindicators have been carried out analysing the HM accumulation in the whole thallus in species of the genus *Cystoseira* s.l. [14,20,26,28,29,30,31], or in different portions of the frond, in species of the genus *Fucus* Linnaeus [36]. Here, for the first time, we differentially analysed the element concentration in the perennial base crust and in the deciduous frond as a system for estimating long-term and short-term HM accumulation, respectively. Indeed, the analysis of different algal portions exhibited a clearly different pattern of accumulation, which was the same for both species of algae. For most of the elements, the basal portion has shown a significantly higher concentration of HMs with respect to the frond, as their accumulation is limited to the growing season in the frond. In addition, the trend of the BAF ratio was different between the thallus portions, with the basal crust exhibiting the following order: Pb > Cr > Ni > Cd > Mn > Cu > Zn, while the frond followed a different order: Cd > Ni > Pb > Cr > Cu > Mn > Zn. This BAF order was consistent between the two species. Elements such as cadmium, lead, chromium, nickel, and mercury have an abiogenic nature and thus have no endogenous mechanism of clearance [37]. This may explain their higher accumulation in the thalli observed here. Furthermore, their higher concentration in the base than in the frond may suggest a slow absorption kinetics of these elements and the need for long-term monitoring systems to assess their presence. On the other hand, the low BAF ratios for Mn, Cu, and Zn are mainly due to their higher amounts detected in the water and to the fact that they are biologically functional elements, so they do not passively accumulate in the algae [38,39]. A low bioaccumulation factor value for Cu and Zn was also found in certain species of *Cystoseira* s.l. by Sales et al. [25], indicating that, as in water, these metals tend to accumulate more in the sediment than in algae. Furthermore, Sales et al. [25] found that As has a higher BAF value, indicating its greater accumulation in the biotic component compared to the abiotic one.

In the present study, the HM concentration in the thalli was notably different from that in the water, especially in samples from the industrial area, revealing a correlation with the anthropogenic source. Among the analysed samples, we observed a clear accumulation pattern, with the basal crust showing a significantly higher number of HMs, suggesting a long-term accumulation process that has more toxic effects on the ecosystem [40]. For this reason, their level should be monitored using reliable means able to evidence variation in the composition across the years. Both of the species of *Cystoseira* s.l. investigated here showed a very similar pattern of HM accumulation in relation to the industrialisation degree of the area. However, *E. mediterranea* revealed a clear differentiation between samples from nonindustrial and industrial areas and between the frond and base algal portions sampled in the industrial area. This can make *E. mediterranea* a more sensitive bioindicator for the monitoring of HMs, even if the narrower distribution of this species can restrict its use to the southern Mediterranean. On the other hand, the similar BAF pattern observed in *C. compressa,* both in the basal crust and in the frond, along with its wider distribution, enables this species to be a promising candidate for biomonitoring purposes when *E. mediterranea* is not present. In addition, the localisation of these two species in the shallow zones of rocky substrates facilitates sampling procedures, dramatically reducing the related operational costs as it is not necessary to perform them with nautical means. Furthermore, this allows for extending the monitoring period and providing greater reliability to the pollution estimate.

In conclusion, in the present study, we described, for the first time, the different accumulation patterns of two *Cystoseira* s.l. species in relation to the anthropogenic pressure and algal portion. The obtained results show consistent HM levels related to environmental conditions and between the perennial basal crust and the deciduous frond. These findings allow us to propose both *C. compressa* and *E. mediterranea* as valuable bioindicators for short- and long-term monitoring of HM levels within the Mediterranean Sea.

## 4. Materials and Methods

### 4.1. Study Area

For this study, we analysed three different sites on the east coast of Sicily, Italy: Penisola Magnisi (PM) (37.14585 N, 15.24311 E), Capo Mulini (CM) (37.57431 N, 15.17347 E), and Brucoli (Br) (37.29385 N, 15.20504 E) (Figure 4). The first site is subjected to various anthropic stresses, mainly due to the several industries that are located in the surrounding area and therefore flow into the area the wastewater resulting from their activities; for this reason, the presence of heavy metals can be found in this area. Differently, in the other two sites, the anthropic impact is lower, and there is no direct inflow of wastewater from large industries. Particularly, the CM site falls within zone “C” of the Isole Ciclopi Marine Protected Area. 

### 4.2. Experimental Design

Thalli of *C. compressa* and *E. mediterranea* were sampled in PM, the highly industrial site, where we expected a higher presence of HMs. These thalli were compared to the ones from the nonindustrial areas (control sites) sampled at CM for *C. compressa* and Br for *E. mediterranea*. Sampling was carried out in 2022 to cover the entire life cycle of the species. In addition, seawater was also sampled at each site and for each sampling period to compare the HMs found there and the corresponding species at each site.

Each sampling was carried out staying on the rock outside the sea water. Approximately 10 thalli were sampled with the use of a pickaxe chosen randomly within the selected population and placed in plastic bags. For the sampling of seawater, three test tubes were used. The samples obtained were kept on ice and transferred to the Laboratory of Phycology of the Department of Biological, Geological, and Environmental Sciences of the University of Catania. Each thallus was cleared of epiphytes using a ceramic knife, plastic tweezers, and brushes. Then, the thalli were washed in distilled water, separated into bases and fronds, and stored along with the seawater samples at −20 °C until analysed. We regarded the basal portion of the algae as the basal crust attached to the substrate from which the cauloids arise and the frond portion as the set of primary and higher-order branches.

### 4.3. Chemicals and Standard Solutions

The following reagents were used for sample pretreatment: ultrapure water, concentrated nitric acid (65%, *v*/*v*), and hydrogen peroxide (30%, *v*/*v*). They were purchased from J.T. Baker (Milan, Italy). Standard solutions of Cr, Cu, Pb, Zn, As, Mn, and Ni (1000 mg/L in 2% nitric acid) were purchased from Fluka (Milan, Italy) and that of Cd (1000 mg/L in 2% nitric acid) from Merck (Darmstadt, Germany). The use of these solutions allowed the construction of seven-point calibration curves (0.2, 1.0, 2.0, 5.0, 10, 20, and 50 μg/L). Standard solutions of Sc, Ge, In, and Bi (1000 mg/L in 2% nitric acid), used as online internal standards at a concentration of 1.5 mg/L, and a standard solution of Re (1000 mg/L in 2% nitric acid), used at a concentration of 0.5 mg/L, were obtained from Fluka (Milan, Italy). The standard solution of Hg (1000 mg/L) used for the Hg analysis was provided by CZECH Metrology Institute Analytika (Prague, Czech Republic).

### 4.4. Samples Preparation

#### 4.4.1. Seawater

Seawater samples were prepared for subsequent ICP-MS analysis in two steps: dilution with 2% nitric acid (1:50 *v*/*v* dilution ratio) and filtration using a syringe and 0.45 µm nylon membrane filters.

#### 4.4.2. Algae

The mineralisation of the algal samples was carried out according to the method proposed by [41] with some modifications. In brief, 1 g of each sample was weighed into PTFE vessels, to which 1 mL of internal Re standard at 0.5 mg/L was added. The reagents used for the digestion were 8 mL of HNO_3_ (65%, *v*/*v*) and 2 mL of H_2_O_2_ (30%, *v*/*v*). Ethos 1 microwave system (Milestone, Bergamo, Italy) was used, and the mineralisation was carried out in two steps: 10 min at 1000 W up to 200 °C and 10 min at 1000 W at 200 °C. After cooling, each sample was made up to 25 mL with ultrapure water. A solution of 1 mL of the internal standard of Re at 0.5 mg L^−1^, 8 mL of HNO_3_, and 2 mL of H_2_O_2_ was used as a blank solution. The samples were filtered through a nylon filter of 0.45 μm to remove the larger particles. The certified reference material (SRM 1570a (Spinach Leaves)) was digested under the same conditions as the samples. All determinations were carried out in triplicate. The samples were filtered using a 0.45 μm filter to remove the larger particles. The certified reference material (SRM 1570a (Spinach Leaves)) was digested under the same conditions as the samples. All determinations were carried out in triplicate.

### 4.5. Analysis by ICP-MS

Cu, Cd, Cr, Pb, Zn, Mn, Ni, and As contents were determined using an iCAP-Q ICP-MS spectrometer (Thermo Scientific, Waltham, MA, USA). The ICP-MS characteristics and operating conditions were reported in the previous work [42]. The monitored isotopes were ^52^Cr, ^55^Mn, ^60^Ni, ^63^Cu, ^66^Zn, ^75^As, ^114^Cd, and ^208^Pb. Integration times were 0.5 s/point for As and 0.1 s/point for other elements. All samples were analysed in triplicate.

### 4.6. Hg Analysis

Hg analysis was performed using DMA-80 (Milestone S.r.l., Sorisole, Italy), a direct mercury analyser of solid, liquid, and gas samples. The DMA-80 was used according to the previous study by [43]. Overall, 0.1 g of each algae sample and 100 µL of each seawater sample were placed in a nickel cuvette and quartz cuvette, respectively, and then subjected to a temperature ramp from 60 °C to 650 °C for approximately 5–6 min. The Hg content was determined by atomic absorption spectroscopy at a wavelength of 253.7 nm.

### 4.7. Validation Parameters

To validate the method, the limit of detection (LOD), the limit of quantification (LOQ), the linearity, and the accuracy (Table 3) were determined according to Eurachem criteria [44]. LODs and LOQs were calculated as 3.3 σ/S and 10 σ/S, respectively, where σ is the standard deviation of six blanks, and S is the slope of the relative calibration curve. The linearity was determined using linear least squares regression and by seven-point calibration curves (range: 0.2–50.0 μg/L).

For the assessment of accuracy for algae samples, the SRM 1570a (Spinach Leaves) was analysed (*n* = 6). The result was reported as mean percentage recovery (%). If the element was not present in the certified matrix, this was spiked with the known amount of the analyte.

Recovery tests for seawater samples were performed in triplicate using ultrapure water samples spiked at three different concentration levels: 0.1, 0.2, and 0.5 mg/kg for Pb, Ni, Cr, As, Cd, and Hg and 0.5, 1.0, and 2.0 mg/kg for Zn and Cu. The results are presented as average recoveries (%) of the three levels.

### 4.8. Data Analysis

The differences in metal concentrations among algae portions and sites were analysed using a two-way ANOVA, followed by post hoc Tukey tests (*p* < 0.05) to compare the metal accumulation between frond and base and industrial (PM) and control sites (Br or CM). When the data did not comply with the parametric assumption of normality, they were normalised using the “box cox” function [45] in the MASS package (v.7.3) [46] in R [47]. In all tests, the significance level for differences in critical values was set at *p* < 0.05.

We used the bioaccumulation factor (BAF), which is the ratio of the amount of an element present in the algae (mg/kg dry weight) to the amount of that element present in the water (mg/kg dry weight), to figure out how well each algae could accumulate each metal. 

To investigate relationships among sites, portions of the thallus, and element concentrations in biota, Principal Component Analysis (PCA) was carried out using the package factorMineR (v. 2.9) [48] in R. The relative contributions were visualised in vector plots.

## Figures and Tables

**Figure 1 plants-13-00530-f001:**
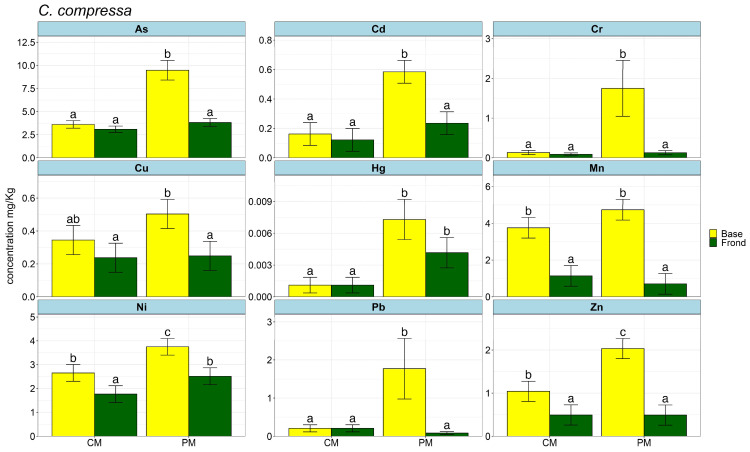
Bar plots of HM mean accumulation across the year in *C. compressa* measured in the base and frond. Significant variation is shown with different letter labels.

**Figure 2 plants-13-00530-f002:**
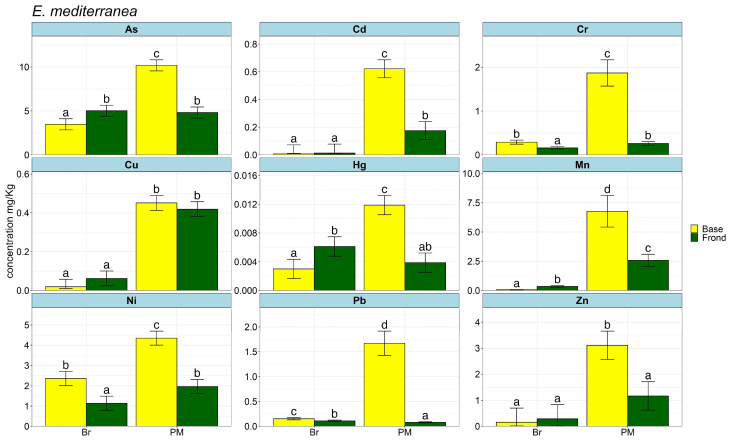
Bar plots of HM mean accumulation across the year in *E. mediterranea* measured in the base and fronds. Significant variation is shown with different letter labels.

**Figure 3 plants-13-00530-f003:**
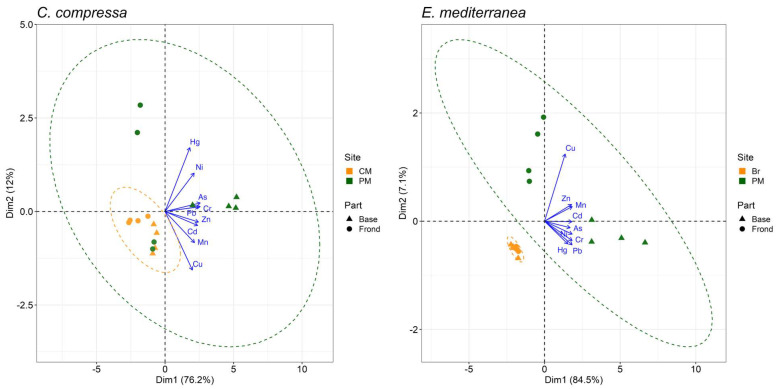
PCA of HM levels over the year in the base and frond of *C. compressa* (**left plot**) and *E. mediterranea* (**right plot**).

**Figure 4 plants-13-00530-f004:**
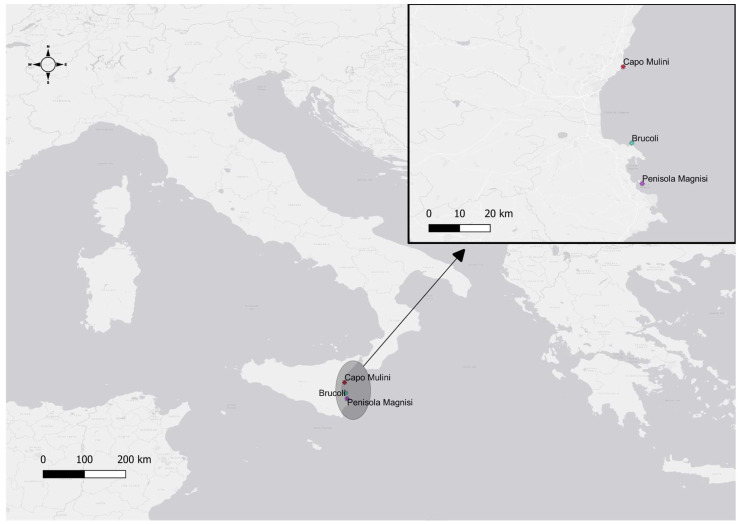
Map of sampling sites: Capo Mulini (CM); Penisola Magnisi (PM); and Brucoli (Br). The map was realised via QGIS software 3.16.

**Table 1 plants-13-00530-t001:** Concentrations (mg/kg ww) of HMs in water samples. (LOQ = limit of quantification).

Site	Sampling Date	Hg	Cu	Cd	Cr	Pb	Zn	As	Mn	Ni
CM	28 January 2022	<LOQ	0.993	0.026	0.010	0.009	2.268	<LOQ	0.985	0.032
	29 April 2022	<LOQ	0.135	0.003	0.006	<LOQ	1.640	<LOQ	0.339	0.018
	19 July 2022	<LOQ	0.243	0.013	0.006	0.012	1.779	<LOQ	0.663	0.019
	13 September 2022	<LOQ	0.328	0.006	0.005	0.008	1.921	<LOQ	0.503	0.012
PM	27 January 2022	<LOQ	0.147	0.021	0.011	0.011	2.096	<LOQ	0.884	0.039
	7 March 2022	<LOQ	0.038	<LOQ	0.007	<LOQ	1.100	<LOQ	0.028	0.020
	8 April 2022	<LOQ	0.057	0.003	0.008	0.002	1.466	<LOQ	0.219	0.021
	28 July 2022	<LOQ	0.076	0.007	0.009	0.005	1.832	<LOQ	0.410	0.023
	3 October 2022	<LOQ	0.093	<LOQ	0.007	0.005	1.776	<LOQ	0.467	0.027
Br	11 March 2022	<LOQ	0.133	<LOQ	0.008	<LOQ	2.292	<LOQ	0.102	0.013
	5 May 2022	<LOQ	0.114	<LOQ	0.010	<LOQ	3.218	<LOQ	0.134	0.019
	22 July 2022	<LOQ	0.102	<LOQ	0.010	<LOQ	2.887	<LOQ	0.114	0.016
	7 September 2022	<LOQ	0.113	<LOQ	0.014	<LOQ	3.121	<LOQ	0.115	0.011

**Table 2 plants-13-00530-t002:** The BAF values calculated in different tissues of *C. compressa* and *E. mediterranea* sampled in PM on 8 April 2022. A complete list of BAFs is provided as Appendix A.

	Hg	Cu	Cd	Cr	Pb	Zn	As	Mn	Ni
*C. compressa* base	-	9.230	159.231	222.188	814.667	1.585	-	24.245	196.833
*C. compressa* frond	-	8.301	124.615	39.438	48.000	0.342	-	5.616	92.286
*E. mediterranea* base	-	7.761	148.769	176.688	643.778	1.337	-	20.683	170.357
*E. mediterranea* frond	-	6.150	80.923	33.438	42.222	0.284	-	8.775	74.167

**Table 3 plants-13-00530-t003:** Analytical validation of ICP-MS and DMA-80 methods of analysis performed in terms of limit of detection (LOD), limit of quantification (LOQ), linearity, and accuracy (n = 6).

Element	LOD (mg/Kg)	LOQ (mg/Kg)	R^2^	SRM 1570a (Spinach Leaves) (%)	Spiked Ultrapure Water (%)
As	0.001	0.003	0.9997	98.00 ± 1.01	97.85 ± 0.95
Cd	0.001	0.003	0.9999	101.00 ± 1.05	101.50 ± 1.12
Cr	0.001	0.003	0.9996	97.35 ± 0.61	97.05 ± 0.50
Cu	0.003	0.010	0.9995	96.00 ± 1.01	96.10 ± 0.98
Hg	0.0003	0.001	0.9998	98.50 ± 1.10	98.35 ± 1.00
Ni	0.003	0.010	0.9996	97.20 ± 0.81	97.42 ± 1.03
Pb	0.0003	0.001	0.9999	105.00 ± 1.06	104.85 ± 1.03
Zn	0.010	0.033	0.9995	98.12 ± 0.77	98.30 ± 0.93

## Data Availability

Not applicable.

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
