# Peer review of "Cystoseira compressa and Ericaria mediterranea: Effective Bioindicators for Heavy- and Semi-Metal Monitoring in Marine Environments with Rocky Substrates"

_plants, 2024, doi:10.3390/plants13040530_

Round 1

Reviewer 1 Report

Comments and Suggestions for Authors

The manuscript presents a comprehensive study on algae as bioindicators for heavy- and semi-metal monitoring in marine environments with rocky substrates. The authors propose the thesis that Cystoseira s.l. species can be used for monitoring heavy metal concentration in areas affected by high levels of industrialisation. They used two common Mediterranean brown macroalgae, Cystoseira compressa and Ericaria mediterranea, as bioindicators to quantify heavy metal pollution in the absence of sediment. They differentially analysed the element concentration in the perennial base and the deciduous frond to estimate the long-term and short-term heavy metal accumulation.

Some grammar and vocabulary mistakes should be corrected. I made some suggestions below, but English is not my first language, so I recommend a revision by someone more competent.

I propose these corrections:

Introduction

According to the journal guidelines, references should be embedded in the text as reference numbers, placed in square brackets [ ], and placed before the punctuation

L 18: change „is not applicable to“ into „does not apply to“

L 24: the algae species showed

L 37: unify industrialisation…industrialization…

L 39: the reference Karaded et al. 2004 is missing in the reference list

L 57/58: correct into Chiarelli and Roccheri, 2014 and Zalewska and Danowska, 2017 (if using the „free format“ at this revision stage)

L 64: delete 2014;

L 68: the reference Pellegrini et al., 1993 is missing in the reference list

L 79: delete space before Guiry

Materials and Methods

L 107: change the term „not-industrialised“

L 140: Hg analysis

Results

L 249/259: the numbers in the brackets are for C. compressa? Please check.

L 255/256: Please check the numbers and sampling stations

L 265: on average lower than 0.01 mg/L

L 266: insert full stop after mg/kg

L: 268: Mn values seem to be higher for C. compressa at PM

L 273: rewrite this part of the sentence to be more clearer

L 274: please check the value 1.44. ± 0.02 mg/kg for C. compressa

L 277: it seems that the accumulation of Zn was higher in the E. mediterranea base of Br samples

Figure 4: correct the site name (CM to Br) in the right plot

L 327/328: on algae as bioindicators

Discussion

I suggest including some references on HM concentrations (other than As) detected in different macroalgae species (L 327, after „Cystoseira complex.“) and also after the L 346.

Also, please include the discussion on the values of BAF for heavy metals in different macroalgae, providing some references for the same.

References

Please use consistent formatting (punctuation, etc.); Latin genus/species names should be italicised.

L 398: 2003 should not be in bold

L 424: the name of the publisher and publisher location are missing…please uniformise all book/book chapter references in the list

L 451: please correct into MSFD

Comments on the Quality of English Language

Some grammar and vocabulary mistakes should be corrected. I made some suggestions below, but English is not my first language, so I recommend a revision by someone more competent.

Introduction

L 18: change „is not applicable to“ into „does not apply to“

L 24: the algae species showed

L 37: unify industrialisation…industrialization…

Materials and Methods

L 107: change the term „not-industrialised“

Results

L 273: rewrite this part of the sentence to be more clearer

L 327/328: on algae as bioindicators

Author Response

The manuscript presents a comprehensive study on algae as bioindicators for heavy- and semi-metal monitoring in marine environments with rocky substrates. The authors propose the thesis that Cystoseira s.l. species can be used for monitoring heavy metal concentration in areas affected by high levels of industrialisation. They used two common Mediterranean brown macroalgae, Cystoseira compressa and Ericaria mediterranea, as bioindicators to quantify heavy metal pollution in the absence of sediment. They differentially analysed the element concentration in the perennial base and the deciduous frond to estimate the long-term and short-term heavy metal accumulation.

Some grammar and vocabulary mistakes should be corrected. I made some suggestions below, but English is not my first language, so I recommend a revision by someone more competent.

I propose these corrections:

Introduction

According to the journal guidelines, references should be embedded in the text as reference numbers, placed in square brackets [ ], and placed before the punctuation

L 18: change „is not applicable to“ into „does not apply to“

Authors’ answer: The change was made accordingly

L 24: the algae species showed

Authors’ answer: The change was made accordingly

L 37: unify industrialisation…industrialization…

Authors’ answer: The changes were made accordingly

L 39: the reference Karaded et al. 2004 is missing in the reference list

Authors’ answer: The references list was amended accordingly

L 57/58: correct into Chiarelli and Roccheri, 2014 and Zalewska and Danowska, 2017 (if using the „free format“ at this revision stage)

Authors’ answer: The changes were made accordingly

L 64: delete 2014;

Authors’ answer: The change was made accordingly

L 68: the reference Pellegrini et al., 1993 is missing in the reference list

Authors’ answer: The references list was amended accordingly

L 79: delete space before Guiry

Authors’ answer: The change was made accordingly

Materials and Methods

L 107: change the term „not-industrialised“

Authors’ answer: The change was made accordingly throughout the manuscript

L 140: Hg analysis

Authors’ answer: The change was made accordingly

Results

L 249/259: the numbers in the brackets are for C. compressa? Please check.

Authors’ answer: We thank the reviewer for his comment. Upon checking the values, we realized that the original submission had utilized incorrect plots. There are no substantial differences with respect to the new plots, as there are only slight changes in the mean values. We have now added the correct plots and adjusted the mean values accordingly in the text. Furthermore, we have added a supplementary table (Table s1) with all the mean values used in the results section.

L 255/256: Please check the numbers and sampling stations

Authors’ answer: this refers to the previous comment.

L 265: on average lower than 0.01 mg/L

Authors’ answer: text was amended as suggested.

L 266: insert full stop after mg/kg

Authors’ answer: The text was amended accordingly.

L: 268: Mn values seem to be higher for C. compressa at PM

Authors’ answer: this refers to the previous comment about the wrong plots. The description is now correct.

L 273: rewrite this part of the sentence to be more clearer

Authors’ answer: The sentence was rewrote accordingly

L 274: please check the value 1.44. ± 0.02 mg/kg for C. compressa

Authors’ answer: this refers to the previous comment about the wrong plots. The description is now correct.

L 277: it seems that the accumulation of Zn was higher in the E. mediterranea base of Br samples

Authors’ answer: this refers to the previous comment about the wrong plots. The description is now correct.

Figure 4: correct the site name (CM to Br) in the right plot

Authors’ answer: The Figure 4 was amended as suggested.

L 327/328: on algae as bioindicators

Authors’ answer: The text was amended accordingly.

Discussion

I suggest including some references on HM concentrations (other than As) detected in different macroalgae species (L 327, after „Cystoseira complex.“) and also after the L 346.

Authors’ answer: We amended the text accordingly.

Also, please include the discussion on the values of BAF for heavy metals in different macroalgae, providing some references for the same.

Authors’ answer: Following reviewer’ suggestion we have added a comparison with other macroalgae species. However, since the thallus structure is substantially different among macroalgae, we limited our comparison to the genus Cystoseira s.l. in order to keep a fair comparison.

References

Please use consistent formatting (punctuation, etc.); Latin genus/species names should be italicised.

Authors’ answer: We amended the reference list accordingly.

L 398: 2003 should not be in bold

Authors’ answer: We amended the text accordingly.

L 424: the name of the publisher and publisher location are missing…please uniformise all book/book chapter references in the list

Authors’ answer: We amended the reference list accordingly.

L 451: please correct into MSFD

Authors’ answer: We amended the text accordingly.

Reviewer 2 Report

Comments and Suggestions for Authors

I am only going to make biological comments on the ms., since the methodology of chemical analysis of heavy metals is not my expertise.

1-Lines 116-118. “Each thallus was cleared of epiphytes using razor blades, tweezers, and brushes. Then the thalli were washed in distilled water, separated into bases and fronds, and stored along with the seawater samples at -20 °C until analyzed.”

Comment: Care must be taken because the use of metallic instruments (razor blades, tweezers) can contaminate the samples with metals. On the other hand, Cystoseira s.l. are hemiphanerophytic algae, with a perennial base and seasonal developing branches. How both parts are delimited should be specified a little better.

2-Lines 338-340. “Elements such as arsenic, cadmium, lead, chromium, nickel, and mercury have an abiogenic nature and thus have no endogenous mechanism of clearance (Zhao et al., 2022)”

Comment: This is not entirely true since the content in As total does not say anything, as already cited in Almela et al. (2002), a paper that is cited in this ms. It is known that especially brown algae of the order Fucales, as is the case, actively accumulate organic As instead of P for its use as a catalyst in different physiological processes, hence it can be surprisingly abundant in them despite being practically undetectable in the marine environment.

3-Lines 328-331: “here, for the first time, we differentially analyzed the element concentration in the perennial base crust and in deciduous frond as a system for estimating the long-term and short-term HMs accumulation, respectively.”

Comment: This statement is not true since there have been works for many years in Fucales that differentiate the accumulation of heavy metals in the basal and apical parts of the thalli by estimating the long-term and short-term HMs accumulation, respectively. For example:

Barreiro, R., Real, C., Carballeira, A. (1993). Heavy-Metal Accumulation by Fucus ceranoides in a Samll Estuary in North-West Spain. Marine Environmental Research 36: 39-61.

4-Lines 356-360 “E. mediterranea revealed a clear differentiation between samples from non-industrialized and industrialized areas and between the frond and base algal portions sampled in the industrialized area. This can make E. mediterranea a more sensitive bioindicator for the monitoring of HMs, even if the narrower distribution of this species can restrict its use to the southern Mediterranean."

Comment: It is not explained why this occurs, but the reason may be that C. compressa lives in the highest areas of the coast and in calm waters, while E. mediterranea lives in environments with more hours underwater and greater hydrodynamism (higher rate of water renewal), which can cause their metal accumulation rates to be higher.

 Other corrections:

Throughout the text Cystoseira s.l. should be replaced. by Cystoseira s.l.

Author Response

I am only going to make biological comments on the ms., since the methodology of chemical analysis of heavy metals is not my expertise.

1-Lines 116-118. “Each thallus was cleared of epiphytes using razor blades, tweezers, and brushes. Then the thalli were washed in distilled water, separated into bases and fronds, and stored along with the seawater samples at -20 °C until analyzed.”

Comment: Care must be taken because the use of metallic instruments (razor blades, tweezers) can contaminate the samples with metals.

Authors’ answer: We thank the reviewer for his remark; we used plastic tweezers and ceramic knives; we have now specified this in the text.

Comment: On the other hand, Cystoseira s.l. are hemiphanerophytic algae, with a perennial base and seasonal developing branches. How both parts are delimited should be specified a little better.

Authors’ answer: We have specified more in the text the methodology we followed to distinguish the 'base' and the 'frond'

2-Lines 338-340. “Elements such as arsenic, cadmium, lead, chromium, nickel, and mercury have an abiogenic nature and thus have no endogenous mechanism of clearance (Zhao et al., 2022)”

Comment: This is not entirely true since the content in As total does not say anything, as already cited in Almela et al. (2002), a paper that is cited in this ms. It is known that especially brown algae of the order Fucales, as is the case, actively accumulate organic As instead of P for its use as a catalyst in different physiological processes, hence it can be surprisingly abundant in them despite being practically undetectable in the marine environment.

Authors’ answer: Thank you for the interesting consideration. We have removed As from the specific sentence and integrated your input into the discussion.

3-Lines 328-331: “here, for the first time, we differentially analyzed the element concentration in the perennial base crust and in deciduous frond as a system for estimating the long-term and short-term HMs accumulation, respectively.”

Comment: This statement is not true since there have been works for many years in Fucales that differentiate the accumulation of heavy metals in the basal and apical parts of the thalli by estimating the long-term and short-term HMs accumulation, respectively. For example:

Barreiro, R., Real, C., Carballeira, A. (1993). Heavy-Metal Accumulation by Fucus ceranoides in a Samll Estuary in North-West Spain. Marine Environmental Research 36: 39-61.

Authors’ answer: This is an interesting observation; however, in the suggested work, the authors consider the different accumulation capacity between the apical portion (which in Fucus is the most distal part of the frond) and the adult portion located 5 cm below the apical portion. This adult portion may have been formed only a few months or at most 1-2 years earlier. While in Cystoseira s.l., we instead consider the perennial crustose portion and the seasonal frond. For this reason, we state that our observations are being reported for the first time.

4-Lines 356-360 “E. mediterranea revealed a clear differentiation between samples from non-industrialized and industrialized areas and between the frond and base algal portions sampled in the industrialized area. This can make E. mediterranea a more sensitive bioindicator for the monitoring of HMs, even if the narrower distribution of this species can restrict its use to the southern Mediterranean."

Comment: It is not explained why this occurs, but the reason may be that C. compressa lives in the highest areas of the coast and in calm waters, while E. mediterranea lives in environments with more hours underwater and greater hydrodynamism (higher rate of water renewal), which can cause their metal accumulation rates to be higher.

Authors’ answer: Thank you for your observation; however, the sites were identified on the basis of the same exposure, coastal height and comparable hydrodynamic rhythms

Other corrections:

Throughout the text Cystoseira s.l. should be replaced by Cystoseira s.l.

Authors’ answer: The change was made accordingly throughout the manuscript

Reviewer 3 Report

Comments and Suggestions for Authors

The research was well conducted and reported, illustrating how these two microalgae can serve as good indicators for metal contamination.

I have a few comments that require attention:

Section 2.1 and Fig. 1.: «Penisola Magnisi (PM) in the territory of Priolo Gargallo (SR)…» These details are not useful to the reader, regarding the ‘territory’ or the abbreviations ‘SR’ or ‘CT’. It is more useful to explain all the sampling were conducted in the east coast of Sicily, Italy. The reader has to guess these coordinates simply belong to the Mediterranean area.

Line 112: «…thalli were sampled with the use of a pick». Details are missing here. Was sampling conducted during low tide, by foot? No need to dive? (line 365 - «it is not necessary to perform them with nautical means»)

Line 144: which type of membrane was used?

Table 1- The results for the SRM should be in % not in mg/kg (line 205: «mean percentage recovery»)

Table 1- same comment for spiked water, recovery in percentage please. What about the 3 levels of spiking? I did not found their individual recoveries. Is this an average recovery of the 3 levels?

Line 312: «The risk of HMs pollution for the protection of biodiversity in the marine environment requires reliable and low-cost bio-indicators»  Rewrite the sentence please, such as ‘For the protection of biodiversity from The risk of HMs pollution in the …’

Comments on the Quality of English Language

none

Author Response

The research was well conducted and reported, illustrating how these two microalgae can serve as good indicators for metal contamination.

I have a few comments that require attention:

Section 2.1 and Fig. 1.: «Penisola Magnisi (PM) in the territory of Priolo Gargallo (SR)…» These details are not useful to the reader, regarding the ‘territory’ or the abbreviations ‘SR’ or ‘CT’. It is more useful to explain all the sampling were conducted in the east coast of Sicily, Italy. The reader has to guess these coordinates simply belong to the Mediterranean area.

Authors’ answer: The change was made accordingly.

Line 112: «…thalli were sampled with the use of a pick». Details are missing here. Was sampling conducted during low tide, by foot? No need to dive? (line 365 - «it is not necessary to perform them with nautical means»)

Authors’ answer: At our latitudes, the tidal range is irrelevant, but we specified it better in the text.

Line 144: which type of membrane was used?

Authors’ answer: Nylon membrane filters were used. We specified it better in the text

Table 1- The results for the SRM should be in % not in mg/kg (line 205: «mean percentage recovery»)

Authors’ answer: The change was made accordingly.

Table 1: same comment for spiked water; recovery in percentage, please. What about the 3 levels of spiking? I did not found their individual recoveries. Is this an average recovery of the 3 levels?

Authors’ answer: The changes were made accordingly.

Line 312: «The risk of HMs pollution for the protection of biodiversity in the marine environment requires reliable and low-cost bio-indicators» Rewrite the sentence please, such as ‘For the protection of biodiversity from The risk of HMs pollution in the …’

Authors’ answer: The sentence was amended as suggested.

Round 2

Reviewer 1 Report

Comments and Suggestions for Authors

 Manuscript could be accepted in present form.